# Unraveling Resistance to Immunotherapy in MSI-High Colorectal Cancer

**DOI:** 10.3390/cancers15205090

**Published:** 2023-10-21

**Authors:** Ronald Heregger, Florian Huemer, Markus Steiner, Alejandra Gonzalez-Martinez, Richard Greil, Lukas Weiss

**Affiliations:** 1Department of Internal Medicine III with Hematology, Medical Oncology, Hemostaseology, Infectiology and Rheumatology, Oncologic Center, Salzburg Cancer Research Institute-Laboratory for Immunological and Molecular Cancer Research (SCRI-LIMCR), Center for Clinical Cancer and Immunology Trials (CCCIT), Paracelsus Medical University, 5020 Salzburg, Austriaf.huemer@salk.at (F.H.); mark.steiner@salk.at (M.S.);; 2Cancer Cluster Salzburg, 5020 Salzburg, Austria

**Keywords:** colorectal cancer, mismatch-repair deficiency, microsatellite instability, immune checkpoint inhibitors, immune evasion, immune escape, resistance to immune checkpoint inhibitors

## Abstract

**Simple Summary:**

Mismatch-repair deficient (dMMR)/microsatellite instability high (MSI-H) cancers encompass a subset of colorectal cancers (CRCs) sensitive to immune checkpoint inhibitors (ICIs). Nevertheless, nearly 30% of patients with dMMR/MSI-H CRC show primary resistance to ICIs, and some develop resistance in the course of disease. In this review, we first explore cells involved in immunogenicity and how intracellular and extracellular factors might influence responses to ICIs. Lastly, we depict uncertainties in the diagnosis of dMMR/MSI-H CRC and outline possible approaches to overcome resistance mechanisms.

**Abstract:**

Colorectal cancer (CRC) is the third most common cancer and the second leading cause of cancer-related deaths. Incidences of early CRC cases are increasing annually in high-income countries, necessitating effective treatment strategies. Immune checkpoint inhibitors (ICIs) have shown significant clinical efficacy in various cancers, including CRC. However, their effectiveness in CRC is limited to patients with mismatch-repair-deficient (dMMR)/microsatellite instability high (MSI-H) disease, which accounts for about 15% of all localized CRC cases and only 3% to 5% of metastatic CRC cases. However, the varied response among patients, with some showing resistance or primary tumor progression, highlights the need for a deeper understanding of the underlying mechanisms. Elements involved in shaping the response to ICIs, such as tumor microenvironment, immune cells, genetic changes, and the influence of gut microbiota, are not fully understood thus far. This review aims to explore potential resistance or immune-evasion mechanisms to ICIs in dMMR/MSI-H CRC and the cell types involved, as well as possible pitfalls in the diagnosis of this particular subtype.

## 1. Introduction

Colorectal cancer (CRC) is a significant global health concern, ranking as the third most common cancer diagnosis and the second leading cause of cancer-related deaths in 2020 [1,2,3,4]. In Europe, it accounts for one-eighth of all cancer diagnoses, making it the second most prevalent tumor type [5]. With incidences of early CRC cases rising by 1% to 4% annually in high-income countries [1], there is an urgent need for effective treatment strategies. Immune checkpoint inhibitors (ICIs)—alone or in combination with chemotherapy—have demonstrated considerable clinical efficacy in a wide range of cancer types and have therefore emerged as a cornerstone of standard treatments in many cancers. The effectiveness of ICIs in CRC is limited to patients with mismatch-repair-deficient (dMMR)/microsatellite instability high (MSI-H) disease. This subtype accounts for about 15% of all CRC localized cases [6,7] and only 3% to 5% of metastatic colorectal cancer (mCRC) cases [8,9]. dMMR/MSI-H CRC can arise due to either sporadic or hereditary causes. One out of eight dMMR/MSI-H CRCs cases are sporadic and occur due to somatic promoter hypermethylation of the MutL homolog 1 (MLH1) gene [10]. Hereditary cases are mainly associated with Lynch syndrome and result from germline mutations in one of the MMR genes, such as MLH1, PMS1 homolog 2 (PMS2), MutS homolog 2 (MSH2), and MutS homolog 6 (MSH6) or a mutation of the EPCAM gene. When compared to their mismatch-repair-proficient(pMMR)/microsatellite stable (MSS) counterpart, dMMR/MSI-H CRCs typically exhibit specific tumor characteristics. These include poor differentiation, a high frequency of BRAF mutations, and a tendency for the primary tumor to be located in the right colon [8,11].

The advent of ICIs has revolutionized the treatment landscape for metastatic dMMR/MSI-H CRC. In the phase III clinical trial KEYNOTE-177 and the phase II trial CHECKMATE-142, disease control rates of 65% using pembrolizumab and 84% using a combination treatment of nivolumab and ipilimumab were observed [12,13]. These results underscore the significant potential of ICIs in managing dMMR/MSI-H mCRC, with the median progression-free survival (PFS) extending to 16.5 months compared to 8.2 months with standard chemotherapy and biological agents. As a result, pembrolizumab has been established as a new standard for first-line therapy in these patients [12], with nivolumab and ipilimumab being considered for second-line therapy [13].

However, despite their remarkable efficacy, a subset of patients (about 30–40% of dMMR/MSI-H mCRC) do not respond to ICI treatment, and about 20–25% of patients are refractory to immune checkpoint blockade [12,13]. The reasons behind this lack of response in some patients remain unclear, with a potential misdiagnosis of dMMR/MSI-H status being one of the considerations. This review aims to delve into the current understanding of potential resistance or immune-evasion mechanisms to ICIs in dMMR/MSI-H CRC and to shed light on the cell types involved.

## 2. Cell Types Involved in Tumor Immune Evasion in dMMR/MSI-H CRC

The tumor environment contains various immune cells, such as CD4+ and CD8+ T cells, natural killer cells, regulatory T cells (Tregs), and myeloid-derived suppressor cells (MDSCs). These cells interact in intricate ways that can either promote or hinder tumor growth. T lymphocytes, particularly CD8+ cytotoxic T cells and CD4+ helper T cells, are crucial in distinguishing self from foreign antigens, with the help of antigen-presenting cells like dendritic cells. However, the activation of these T cells can be negatively regulated by immune checkpoint molecules such as cytotoxic T lymphocyte antigen 4 (CTLA-4), programmed cell death protein 1 (PD-1), T-cell immunoglobulin and mucin-domain containing-3 (TIM-3), and lymphocyte activation gene 3 (LAG-3) [14]. This regulation can lead to peripheral tolerance and T-cell exhaustion, reducing the effectiveness of the immune response against the tumor.

### 2.1. Regulatory T Cells

Tregs play a key role in maintaining immune homeostasis by limiting the inflammation caused by effector T cells [15,16]. They achieve this through the release of inhibitory cytokines like interleukin (IL) 10 and transforming growth factor β (TGF-β), or by modulating antigen-presenting cells through the expression of CTLA-4 or LAG-3 [17,18]. However, their role in CRC is still unclear due to conflicting findings on their prognostic significance. For instance, Salama et al. found that the presence of Tregs in CRC was associated with improved survival outcomes [19,20]. On the contrary, Waniczek et al. reported that high Treg infiltration was linked to poor disease-free survival (DFS) and overall survival (OS) [21]. Although these results have been obtained primarily in tumors presumed to be pMMR/MSS, they shed a light on the diversity of Tregs, which include both activated and non-suppressive subtypes.

These subtypes consist of suppressive cluster of differentiation (CD) 45RA+ forkhead-box-protein P3 (FOXP3) high naive-like cells, suppressive CD45RA–FOXP3-high effector Treg cells, and pro-inflammatory CD45RA–FOXP3-low effector Treg cells [22]. Saito et al. found that high FOXP3 expression was linked to poorer prognosis in CRC patients, with tumor-infiltrating Tregs primarily being suppressive effector Tregs (CD45RA–FOXP3-high) [23]. Targeting these suppressive Tregs or their inhibitory cytokines might be a strategy to overcome resistance to ICIs. For instance, antibodies against IL-10 have been shown to increase the presence of tumor-infiltrating lymphocytes (TILs) and promote tumor cell death in vitro [24]. While earlier studies have noted an increase in FOXP3 positive cells in dMMR/MSI-H CRC [25,26], the specific function of these Tregs in facilitating immune evasion in MSI-H CRC tumors is yet to be fully understood.

### 2.2. Myeloid-Derived Suppressor Cells

Myeloid-derived suppressor cells (MDSCs) are a heterogeneous population of immune cells from the myeloid lineage and can suppress anti-tumor immunity through different mechanisms [27]:Enzyme Production: MDSCs produce enzymes such as nitric oxide synthase and arginase-1, which deplete L-arginine, an amino acid that is essential for the normal functioning of T cells and the T-cell receptor [28,29]. This depletion impairs the immune response against tumors.Interaction with Tregs: there is evidence to suggest that MDSCs can stimulate the activation of Tregs through the release of the cytokine IL-10 [30], which leads to immunosuppression.

Even though there is a higher prevalence of intratumoral Tregs and MDSCs in pMMR/MSS CRC compared to dMMR/MSI CRC [31], dMMR/MSI tumors might be more susceptible to the effects of these immunosuppressive cells.

### 2.3. Tumor-Associated Macrophages

Tumor-associated macrophages (TAMs) are generally found in the microenvironment of solid tumors and can be classified into M1 (proinflammatory) and M2 (anti-inflammatory) subtypes [32]. M1 macrophages, typically activated by interferon-γ (IFN-γ) or tumor necrosis factors, possess cytotoxic effects against cancer cells [33,34]. IL-4 is one of several cytokines responsible for activating M2 macrophages, which exhibit anti-inflammatory and pro-tumor characteristics [35]. This polarization state is not permanent, allowing macrophages to switch between the M1 and M2 phenotypes [35]. The role of these macrophages in CRC is debated, with some studies linking high TAM density to improved outcomes [36,37] and others suggesting the opposite [38].

Specific studies on dMMR/MSI-H CRC tumors have shown a higher prevalence of intratumoral M1 macrophages [39] and elevated expression levels of PD-L1 on M2 macrophages at the invasive front [40] in comparison to pMMR/MSS tumors.

To date, three strategies have been explored to target TAMs as a potential cancer treatment: inhibiting TAM recruitment, reprogramming pro-tumoral M2 macrophages to the anti-tumoral M1 phenotype, or a combination of both [41,42].

## 3. Mechanisms of Immune Evasion and Resistance to ICIs

The initial immune response to tumor antigens by the innate and adaptive immune system is termed the elimination phase. This is followed by the equilibrium phase, which describes a state of balance between the persistence of tumor cells and their destruction by the immune system [43]. A possible mechanism of immunologic escape can include changes in the antigen-presenting machinery (APM), the development of an immune-tolerant tumor microenvironment, and the upregulation of immune checkpoint molecules.

### 3.1. Tumor Intrinsic Factors Related to Immune Evasion and ICI Resistance

#### 3.1.1. Antigen-Presenting Machinery

Within the APM, MHC class I molecules are pivotal, as they facilitate T-cell antigen recognition. They are displayed on the surface of almost all nucleated cells and present peptides—e.g., tumor antigens—to T cells [44]. The MHC class I protein is composed of the heavy chain encoded by the human leukocyte antigen (HLA) genes A, B, or C on chromosome 6 and the non-polymorphic light chain encoded by the beta2-microglobulin (β2M) gene situated on chromosome 15 [44,45]. Genetic changes, including mutations and loss of heterozygosity (Table 1 and Figure 1(C1)), as well as disturbances in the assembly (Figure 1(C2)) [46] or breakdown of the MHC complex, can result in its diminished expression on the cell surface (Figure 1(C3)). This may allow tumor cells to evade detection by the immune system (Figure 1(C4)) which contributes to tumor proliferation and migration (Figure 1(C5)) [37]. Alterations or loss of heterozygosity in β2M have been described as possible mechanisms of (acquired) resistance to ICIs in melanoma [44,47] and lung cancer [48]. However, the evidence in dMMR/MSI-H CRC is more ambiguous: even though resistance to ICIs has been linked to β2M alterations in some cases [49], β2M mutations or the absent expression of β2M do not necessarily preclude a response to ICIs [50].

In dMMR/MSI-H CRC, genetic alterations of the APM are commonly observed: 11% of untreated primary tumor samples show a loss of the β2M gene, and 55% exhibit alterations within the APM [51]. In dMMR/MSI-H CRC cases, around 20% have mutations in HLA-A, -B, or -C [52]. This percentage rises to 50% in Lynch syndrome cases [53,54]. However, the elevated mutation rate in these genes might be attributed to the inherent hypermutational status of dMMR/MSI-H cancer [55,56].

In the KEYNOTE 177 trial, patients with RAS-mutated diseases who received ICI treatment with the anti-PD-1 antibody pembrolizumab had a shorter progression-free survival (PFS) when compared to patients with RAS wildtype disease [12]. This observation aligns with a study by Salem et al., which demonstrated that dMMR/MSI-H CRCs with RAS mutations weakened immune surveillance and made the tumor environment less advantageous after ICI treatment [57]. Preclinical research suggests that a KRAS mutation can suppress the expression of HLA-A, -B, or -C by inhibiting interferon regulatory factor 2. This leads to an increased expression of chemokine ligand 3 (CXCL3), prompting immunosuppressive MDSCs to infiltrate the tumor environment [58,59]. Additionally, KRAS mutations might promote an immunosuppressive environment through the activation of the MAPK (Raf/MEK/ERK) pathway [60]. Of interest, the effectiveness of combined immunotherapy using the anti-PD-1 antibody nivolumab and the anti-CTLA-4 antibody ipilimumab remained consistent, irrespective of the presence of a RAS mutation [61].

#### 3.1.2. WNT/β-Catenin Signaling Pathway

The adenomatous polyposis coli (APC) gene, a tumor suppressor, plays a pivotal role in regulating the WNT/β-catenin signaling pathway. One of the key actions of the adenomatous polyposis gene protein is forming a complex with other proteins to target β-catenin for destruction, ensuring its levels remain low (Figure 1A). However, a biallelic loss of the APC gene, as seen in 62% of MSS cases and 20% of MSI-H disease cases [55], leads to the accumulation of β-catenin and the subsequent activation of the WNT signaling pathway. In various tumor types [62,63,64] as well as in CRC, increased beta-catenin levels have been linked to altered T-cell responses and a notable reduction in tumor-infiltrating lymphocytes (TILs), regardless of their microsatellite status [55]. Therefore, an increased WNT/β-catenin signaling pathway may lead to reduced sensitivity to ICIs in dMMR/MSI-H CRC.

#### 3.1.3. Interferon-γ Signaling

IFN-γ plays a pivotal role in anti-tumor immunity by promoting the apoptosis of tumor cells and enhancing the expression of antigen-presenting molecules like MHC-I [65,66].

Janus Kinases (JAKs) are crucial for IFN-γ signaling, and mutations in JAK may lead to reduced tumor cell apoptosis in response to IFN-γ [67] and the reduced expression of MHC-I (Figure 1B). Frameshift mutations in JAK1 have been described for various MSI-H tumors, including CRC [68,69], with the loss of IFN-γ-mediated anti-tumor immune response. Additionally, mutations in JAK, alongside other mutations in the IFN-γ signaling pathway, have been linked to a loss of MHC-I, resulting in resistance to ICI treatment [70,71] and the heightened risk of recurrence following ICI treatment [44].

#### 3.1.4. The Transforming Growth Factor Beta (TGF-β)-Dependent Stromal Subset

TGF-β is a cytokine with roles in a wide array of cellular processes. Elevated levels of TGF-β are frequently observed in tumor cells [72], and increased TGF-β levels have been shown to stimulate growth in the stromal cell components of the tumor microenvironment [73]. Furthermore, high amounts of TGF-β in the tumor microenvironment might induce T-cell exhaustion (Figure 1D) [74] and have been correlated with a negative prognosis in dMMR/MSI-H CRCs [75]. TGF-β may also induce resistance to ICI treatment in dMMR/MSI-H CRC, primarily due to TGF-β-mediated T-cell inhibition [74,76,77,78]. Mutations in the TGF-β receptor type II or its downstream target, SMAD4, can disrupt the negative feedback mechanism, thereby leading to increased TGF-β levels. Alterations in stromal TGF-β signaling have been associated with decreased efficacy of ICIs in preclinical investigations [74,76,79]. SMAD4 mutations are an adverse prognostic indicator in CRC in general [80,81,82,83], and decreased SMAD4 protein expression—as an indicator of SMAD4 mutation—has been documented in 6 to 14% of MSI-H CRCs [82,84,85].

Currently, multiple phase I/II clinical trials aim to target TGF-β with the goal of enhancing or establishing responsiveness to ICI therapy. (ClinicalTrials.gov references: NCT02423343, NCT02517398, and NCT02947165). 

#### 3.1.5. Clinical, Histopathological, and Molecular Variations in dMMR/MSI-H CRCs

dMMR/MSI-H CRC possess a unique clinical, pathological, and molecular profile. They are often linked with right-sidedness, mucinous histology, poor differentiation, and a high frequency of BRAF mutations. Additionally, they are prone to metastasize to distant lymph nodes and cause peritoneal carcinomatosis [8,86]. 

CRC can be classified into four consensus molecular subtypes (CMS) based on their different gene expression: CMS1 (MSI-H-like or immune, ~14%), CMS2 (canonical, ~37%), CMS3 (metabolic, ~13%), and CMS4 (mesenchymal, ~23%) [87].

Most MSI-H tumors predominantly align with the CMS1 category, characterized by a high mutational load, an immunogenic tumor environment, the presence of specific TILs (e.g., CD8+ cytotoxic T lymphocytes, CD4+ T helper 1 cells and natural killer cells), and an excellent prognosis in early stage CRC [37]. Less frequently, MSI-H tumors can fit into the CMS3 category (16%), which is more prevalent in tumors with KRAS mutations, leading to a sequential activation of metabolic pathways. This category has fewer hypermutations, gene hypermethylation, and less immune infiltration compared to CMS1 [60]. Some MSI-H tumors can also be classified as CMS4, known for the poorest prognosis and a TGF-β-rich, immunosuppressive microenvironment (e.g., chronic inflammation and activation of the innate immune system) [75]. These diverse molecular profiles of dMMR/MSI-H tumors could also account for the variable efficacy of ICIs.

### 3.2. Extrinsic Factors Leading to Immune Evasion and ICI Resistance

#### 3.2.1. Variable Expression of Immune Checkpoints

Tumor cells may express immune checkpoints on their surface, which allows them to evade the adaptive immune system. Permanent activation of immune checkpoints causes T-cell exhaustion, eventually leading to immune evasion. Gene expression analysis in dMMR/MSI-H CRC demonstrated high levels of the immune checkpoints CTLA-4, PD-1, PD-L1, LAG-3, and IDO in different compartments, such as tumor-infiltrating lymphocytes, tumor stroma, and the invasive front of the tumor [26]. One notable difference between MSI-H CRC and other highly ICI-responsive cancers such as melanoma or lung cancer is the lower expression of PD-L1 on the tumor cells [88]. One may hypothesize that the lower expression levels of immune checkpoints may confer decreased sensitivity to ICI treatment.

#### 3.2.2. Gut Microbiota

The connection between gut microbiota and the onset, progression, and outcomes of cancer is well established. In CRC, most data evolve around Fusobacterium nucleatum [89], which is more frequently found in MSI-H- or BRAF-mutated cases [90]. Of interest, the presence of Fusobacterium nucleatum in MSI-high CRC is associated with a lower number of TILs [91].

In recent years, several preclinical models [92] and patient cohorts [92,93,94] shed light on the influence of the gut microbiome on therapeutic responses to ICIs and have identified gut-bacterial dysbiosis as a putative mechanism of primary resistance to ICI treatment. The application of antibiotics in temporal proximity to the start of ICI therapy contributes to an abnormal gut microbiome composition [92] and results in worse clinical outcome in various types of cancer [92,93,94]. Fecal microbiota transplantation from melanoma patients responding to ICIs to patients with primary resistance to ICIs is able to induce responses and/or long-lasting disease stabilization upon ICI rechallenge [95,96]. The latter therapeutic approach is currently investigated in dMMR/MSI-H mCRC patients with primary resistance to ICI therapy in a phase II clinical trial (NCT04729322). However, it still must be determined if the abnormal gut microbiota plays a pivotal role in the immune escape of dMMR/MSI-high CRC.

#### 3.2.3. Immunoscore

Tumor cells, host immune cells, and tumor stroma interact between each other and create a distinguished immune signature in various malignancies [97]. As mentioned before, dMMR/MSI-H CRCs have a high amount of tumor-infiltrating lymphocytes (TILs), particularly consisting of CD8+ cytotoxic T cells with potential antitumor activity [98,99] and T helper cells with IFN-γ secretion [100]. A high density of cytotoxic T lymphocytes in CRCs are associated with better relapse-free and overall survival [101,102,103]. This led to the development of the Immunoscore© (IS), which quantifies the density of CD3+ and CD8+ lymphocytes in the tumor center and at its invasive margin (range: 0–4) [104]. The IS represents an independent prognostic factor concerning improved disease-free survival and overall survival [105]. In MSI-H CRC, 56% of cases show very high levels of TILs according to IS, in contrast to 26% of MSS CRC cases [106].A study in patients with dMMR/MSI-H CRC showed that a higher IS was associated with better response to ICI therapy with pembrolizumab [107]. As described above, various molecular mechanisms—such as increased WNT/β-catenin signaling—might lead to decreased TILs [108] and therefore might be predictors of resistance to ICI therapy.

## 4. Uncertainties in dMMR/MSI-H Diagnosis

### 4.1. Discordance between Diagnostic Methods

Two different methods are mainly used to define MMR status: immunohistochemistry (IHC) staining to investigate MMR protein expression on tumor tissue and MSI-H testing on tumor DNA. MLH1, MSH2, MSH6 (MutS homolog 6), and PMS2 (postmeiotic segregation 2) are the four major proteins of the MMR system. IHC should encompass antibodies against these four proteins [88]. Technical errors in pre-analytical procedures such as tissue fixation might result in false negative IHC staining [109], whereas a missense mutation in one of the four MMR genes might lead to a loss of function with preserved protein expression and detectability by IHC, causing false positive results [110]. To date, both IHC and molecular analysis depend on samples gained via tissue-biopsy [88]. IHC is usually performed by a pathologist and highly depends on one’s experience and skills regarding the staining processes and its interpretation. Sensitivity and specificity range from 81% to 100% and from 80% to 92% [111].

There are two microsatellite marker panels using PCR-based molecular testing, the Bethesda and the pentaplex, for the detection of dMMR in CRC. Both work with the two microsatellite markers BAT-25 and BAT-26. The Bethesda panel includes the microsatellite markers D5S346, D2S1123, and D17S250 and has a sensitivity from 67% to 100% and a specificity from 61% to 92%, whereas the pentaplex panel’s sensitivity and specificity, with its additional microsatellite markers NR-27, NR-21, and NR-24, varies from 89% to 100% and from 79% to 100% [112,113].

Another molecular test for analyzing MSI-H is next-generation sequencing (NGS) [114,115]. Besides MSI status, NGS analysis is capable of evaluating other routinely assessed, treatment-relevant biomarkers in CRC, like RAS and BRAF status as well as tumor mutational burden (TMB).

A comparison of results from IHC- and PCR-based test systems in CRC revealed a discrepancy ranging from 1% to 10% [115,116,117]. The sources of errors are the misinterpretation of IHC/molecular DNA testing because of the insufficient amount of tumor cells in the biopsy, variations in staining quality, and the lack of experience of the pathologist [110,118,119]. False normal staining results of IHC may occur by the expression of a non-functional but antibody-binding MMR protein (e.g., MLH1 mutation) [110]. Isolated loss of MSH6 expression with MSS is also a common phenomenon, resulting in a false positive result [120]. A rare microsatellite polymorphism, as seen in African ethnicities, may deliver false positive results of molecular testing [121].

Clinical practice guidelines are as follows: the European Society for Medical Oncology (ESMO) recommends using IHC in the first place, followed by molecular analysis in case of clinical doubt, although, if feasible, simultaneous testing is preferred. The pentaplex-panel is preferred for PCR-based analysis because of its higher sensitivity and specificity. NGS takes a special position within MSI-H testing, given its advantage to assess further information, but has a lack of practicability: NGS is still a time- and resource-consuming procedure and requires a sufficient size of a tumor sample that sometimes has to be acquired by a rebiopsy [88]. For the National Comprehensive Cancer Network (NCCN), a PCR-based confirmation of the MSI-H result on IHC is obligatory.

### 4.2. Intratumoral and Intertumoral Heterogeneity May Contribute to Therapy Resistance

Intratumoral heterogeneity refers to the presence of cells with varied molecular characteristics within a single tumor. By contrast, differences between a primary tumor and its metastases in the same patient are termed intertumoral intrapatient heterogeneity [122,123]. The heterogeneity of an MSI status is rarely observed in CRC: in a study of 369 patients with CRC, 9 out of 48 cases with primary tumors classified as MSI-H (n = 48) had MSS metastases, especially of the peritoneum and ovary [124].

### 4.3. Lynch Syndrome versus Sporadic MSI-H

Lynch syndrome or hereditary non-polyposis colorectal cancer (HNPCC) is an inherited disorder caused by a germline mutation in one of the above-mentioned MMR genes or a mutation of the EPCAM gene, leading to a loss of expression of MSH2. It is an autosomal dominant disease and accounts for approximately 3% of all CRCs [125]. In the rare case of CRC with biallelic germline mismatch repair (MMR) mutations, the syndrome is referred to as constitutional MMR deficiency [126]. On the other hand, the slightly more common “Lynch-like syndrome” is caused either by biallelic somatic MMR mutations or germline alterations in other genes affecting the MMR system (Table 2) [127].

To date, three prospective, multicenter studies have analyzed the data of 74 patients with Lynch Syndrome-associated CRC, who were receiving ICIs, and observed an overall response rate (ORR) between 46 and 71%. Le et al., did not demonstrate a significant difference in ORR between Lynch syndrome and sporadic dMMR/MSI-H CRC. The CHECKMATE-142 study by Overman et al. observed an ORR of 71% for patients with Lynch syndrome compared to 48% in sporadic dMMR/MSI CRC [128,129]. In a recent study by Chalabi et al., treatment with neoadjuvant immunotherapy in dMMR/MSI CRC resulted in better pathologic complete response rates for patients with Lynch syndrome (78%) than in sporadic dMMR/MSI CRC (58%) [130]. This improved PFS for patients with ICI-treated Lynch syndrome was also observed in recent cohort studies [131,132]. A possible explanation for the survival benefit, aside from the younger age of the patients with Lynch syndrome, might be a higher accumulation of somatic mutations and neoantigens, which is subsequently assumed to produce stronger immunoreactions [133].

## 5. Conclusions and Prospect

The introduction of immune checkpoint inhibitors (ICIs) has transformed the therapeutic landscape for metastatic dMMR/MSI-H CRC, offering promising disease control rates and progression-free survival. However, the heterogeneity in the responses, with a subset of patients showing resistance or primary tumor progression, underscores the need for a deeper understanding of the underlying mechanisms. Factors such as the tumor microenvironment, the role of various immune cells, genetic alterations, and the influence of gut microbiota play pivotal roles in modulating the response to ICIs. Additionally, the potential misdiagnosis of or uncertainties about dMMR/MSI-H status, intratumoral and intertumoral heterogeneity, and the distinction between Lynch syndrome and sporadic MSI-H CRC pose challenges in the clinical management of these patients.

Findings from translational studies may help uncover the potential biological mechanisms that cause resistance to immune checkpoint inhibitors in MSI-H colorectal cancers. For example, Fu et al. demonstrated that the application of a STING agonist cancer vaccine in mice led to an upregulation of PD-L1 [134]. Furthermore, its intratumoral application also resulted in the suppression of tumor growth in CRC [135]. In another study conducted on mice, the combined use of a stimulator of interferon gene (STING) agonist and an indoleamine 2,3-dioxygenase (IDO) inhibitor led to the recruitment of CD8+ T cells and dendritic cells, which in turn inhibited tumor cell growth [136]. Additionally, the introduction of a VEGFR2 antibody to a STING agonist demonstrated a synergistic antineoplastic effect against the murine CT26 colon carcinoma line [137]. Therefore, the combined use of a STING agonist and an ICI may potentially enhance antitumor effects.

As mentioned above, high levels of TGF-β in the tumor microenvironment could potentially induce T-cell exhaustion and exacerbate phenotypical changes in T helper 1 cells [74]. The secretion of TGF-β in the tumor microenvironment could potentially lead to resistance to ICIs in dMMR/MSI-H CRC. This is based on preclinical findings that suggest that the combination of ICIs with TGF-β inhibition could be effective [76,77,78]. The integration of preclinical research with a clinical understanding could potentially offer additional insights into uncovering the mechanisms of resistance to ICIs in dMMR/MSI-H CRC.

To optimize therapeutic outcomes, future research should focus on elucidating the intricate interplay between these factors and their collective impact on ICI efficacy. Personalized treatment strategies, informed by a comprehensive understanding of tumor genetics, immune profiles, and patient-specific factors, will be crucial. The advent of next-generation sequencing and other advanced diagnostic tools offers hope for more precise patient stratification and the development of combination therapies that can overcome resistance and maximize the therapeutic potential of ICIs in CRC (Table 3). 

## Figures and Tables

**Figure 1 cancers-15-05090-f001:**
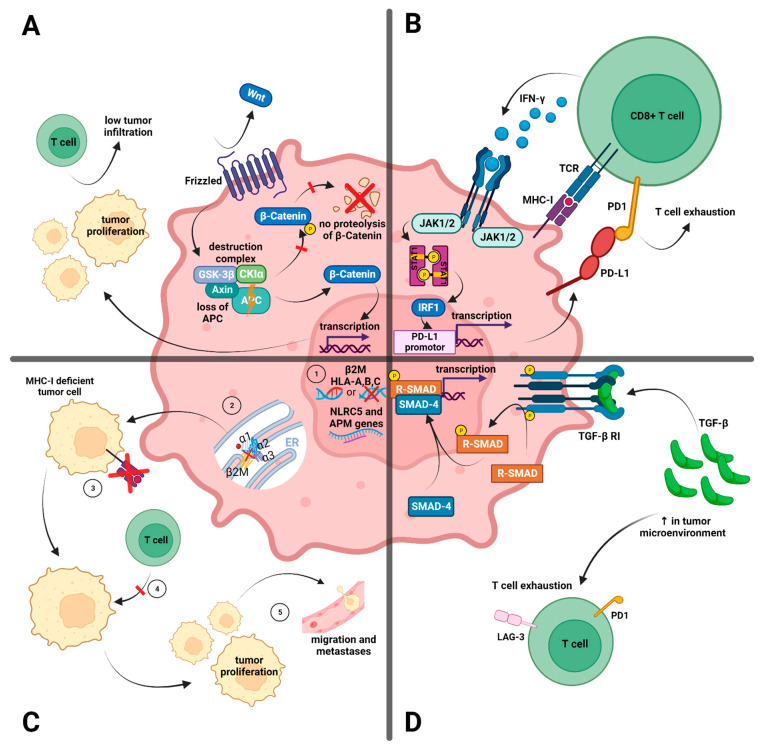
Overview of tumor intrinsic factors related to immune evasion and ICI resistance. (**A**) WNT/β-catenin signaling pathway: the loss of the adenomatous polyposis coli (APC) gene leads to increased β-catenin levels and activation of the WNT signaling pathway, which subsequently results in altered T-cell responses, a significant decrease in tumor-infiltrating lymphocytes, and tumor proliferation; (**B**) Interferon-γ (IFN-γ) signaling: IFN-γ facilitates tumor cell apoptosis and enhances the expression of the major histocompatibility complex class I (MHC-I) and programmed cell death ligand 1 (PD-L1) through the action of Janus Kinases (JAKs). The latter is essential for IFN-γ signaling, and mutations in JAK can lead to reduced tumor cell apoptosis in response to IFN-γ, the diminished expression of MHC-I, and T-cell exhaustion due to PD-L1 overexpression; (**C**) Antigen-presenting machinery (APM): (**1**) intra-tumor mechanisms such as mutations, loss of heterozygosis (LOS), or deletions in the β2M gene, as well as in the HLA-A, -B, and -C genes, trigger the loss of MHC-I. (**2**) Likewise, the improper assembling of β2M and alpha chains inside the endoplasmic reticulum lead to the unsuccessful expression of MHC-I. (**3**) In those situations, MHC-I-deficient tumor cells are not recognized by CD8+ T cells. (**4**) This results in an escape from immunological recognition. (**5**) This opens the way for proliferation and migration to different tissues; (**D**) Transforming growth factor beta (TGF-β): Mutations in either the TGF-β receptor or its downstream target, SMAD4, can result in a surge of TGF-β levels. Elevated levels of TGF-β could lead to the exhaustion of T cells and induce resistance to ICI treatment (created with BioRender.com).

**Table 1 cancers-15-05090-t001:** Overview of HLA-I alterations in tumor cells. HLA-I expression loss depends on the kind of defect: genetic versus non-genetic. Cancer-related alterations can either lead to a complete loss or downregulation of HLA-I expression.

Genetic Defects (“Hard” Lesions)		Non-Genetic Defects (“Soft-Lesions”)
Mutations	LOH at	Transcriptional downregulation: HLA-I genes
HLA-I heavy chain	chr 6	
B2M		
IFN pathway	chr 15	β2M genes IFN pathway APM genes
APM genes		

Legend: HLA = human leucocyte antigen, β2M = beta-2-microglobulin, IFN = interferon, APM = antigen presentation machinery, LOH = loss of heterozygosity, chr = chromosome.

**Table 2 cancers-15-05090-t002:** DNA mismatch repair defects within hereditary and sporadic colorectal cancers.

Mutation Type	Lynch Syndrome	CMMRD	Lynch-like Syndrome
Germline	One allele of MMR gene	Both alleles of MMR gene	None
Somatic	Second allele of MMR gene	None	Both alleles of MMR gene

Legend: CMMRD = Constitutional mismatch repair deficiency.

**Table 3 cancers-15-05090-t003:** Overview of ongoing and recruiting clinical trials of combination therapy with CPI in dMMR/MSI mCRC.

Target	Regimen	Phase	Setting	Identifier
ICI combinations or refractory to first ICI	Nivolumab/Ipilimumab	II	ICI resistant	NCT05310643
	Nivolumab/Ipilimumab	III	ICI naive	NCT04008030
	Nivolumab/Ipilimumab	II	ICI naive	NCT04730544
	IBI310 (anti-CTLA-4)/Sintilimab	II	ICI naive	NCT04258111
	Pembrolizumab +/− Quavonlimab +/− Favezelimab	II	n.a.	NCT04895722
	+/− Vibostolimab +/− MK-4830 (anti-ILT4)			
	Cadonilimab	I/II	ICI resistant	NCT05426005
ICIs plus novel agents	Pembrolizumab/Encorafenib			
	M7824 (anti-PD-L1/TGF-β trap fusion protein)	II	ICI naive	NCT05217446
		I/II	n.a.	NCT03436563
	Pembrolizumab + NC410 (LAIR-2 Fc protein)			
		I/II	n.a.	NCT05572684
	Pembrolizumab + ATRC-101 (anti-RNP)			
	N-803 (IL-15 superagonist) +/− pembrolizumab	I	n.a.	NCT04244552
	+/− nivolumab +/− atezolizumab +/− durvalumab	II	ICI resistant	NCT03228667
	+/− avelumab, respectively			
	Tiselizumab + KFA115 (immunomodulatory agent)	I	ICI naive	NCT05544929
ICIs plus cytotoxic and anti VEGF agents	Pembrolizumab + bevacizumab + FOLFIRI	II	ICI naive	NCT05035381
	Atezolizumab + bevacizumab + FOLFOX	III	ICI naive	NCT02997228
	Toripalimab + bevacizumab + irinotecan	I/II	ICI naive	NCT04988191
	Toripalimab + oxaliplatin + capecitabine	II	ICI naive	NCT04301557
	Camrelizumab + apatinib	II	ICI naive	NCT04715633
ICIs plus radiotherapy	Sintilimab + RT	I/II	ICI naive	NCT04636008
	Nivolumab + ipilimumab + RT	II	ICI naive	NCT03104439
ICI plus COX inhibitor	Toripalimab + celecoxib	I/II	ICI naive	NCT03926338

Legend: ICI = immune checkpoint inhibitor, anti-CTLA-4 = anti cytotoxic T lymphocyte antigen 4, n.a. = not applicable, anti-ILT4 = anti immunoglobulin-like transcript 4, anti-PD-L1 = anti programmed death ligand 1, TGF-β = tumor growth factor beta, LAIR-2 = leukocyte-associated immunoglobulin-like receptor 2, anti-RNP = anti ribonucleoprotein, IL-15 = interleukin 15, VEGF = vascular endothelial growth factor, RT = radiotherapy.

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
