# Peer review of "Unraveling Resistance to Immunotherapy in MSI-High Colorectal Cancer"

_cancers, 2023, doi:10.3390/cancers15205090_

Round 1

Reviewer 1 Report

Review paper exploring mechanisms through which colorectal tumors with the mutator phenotype (MSI-H, MMR Def) can be or become resistant to immunotherapy. Extensive review including pittfalls in diagnosing  this type of tumors, role of tumor microenvironment, namely type of cells and discrepant results of several studies undertaken and exploring reasons for these tumors not responding to ICI. Role of microbiota, previous antibiotic therapy and potential use of fecal transplantation are also reviewed. In this sense I think that it is a complete and useful paper for researchers wanting to develop work in this area. I don’t have any corrections to suggest.

Author Response

Thank you very much for taking the time to review this manuscript.

Reviewer 2 Report

This is a well-written review article regarding the mechanisms of resistance to immunotherapy in MSI-high CRC. However, we would like to know the association between the resistance and location (right- or left-sided) and hisptopathological characteristics (histological type or degree of differentiation) of CRC. If possible, they can be described in a new paragraph. In addition, please complete the references, as some are incomplete. 

Author Response

We thank the reviewer for the valuable input, which has definitely helped to improve our manuscript. Changes have been highlighted in green in the revised manuscript version. Please find below a point-by-point response:

Association between the resistance and location (right- or left-sided), histological type and degree of differentiation:

  • As of now, we have not been able to find data that provides a subgroup analysis for these questions. Neither the Keynote 177 study nor the Checkmate 142 study, which formed the basis for the approval of checkpoint inhibitors in MSI-H CRC, provide this information. In the Keynote 177 study, the hazard ratio for overall survival (pembrolizumab vs chemotherapy), as per a subgroup analysis of sidedness, was 0.72 for right-sided tumors and 0.8 for left-sided tumors, respectively. Nevertheless we added an additional paragraph highlighting clinical and histopathological characteristics of MSI-H CRC (see 3.1.5.).

Please complete the references:

  • It was done accordingly.

Reviewer 3 Report

The authors of the submitted manuscript summarized the current knowledge of the mechanisms responsible for the resistance to immunotherapy in cases of MSI-high colorectal cancer. The manuscript is well-written and provides concise information with well-presented mechanisms in informative figure. I suggest minor modifications.

  1. The authors should provide a more detailed explanation of the molecular subtypes of CRC.
  2. Are there any translational animal models used to study ICI's resistance and to test new approaches to overcome the resistance above? If yes, please provide the appropriate section.

Author Response

We thank the reviewer for the valuable input, which has definitely helped to improve our manuscript. Changes have been highlighted in yellow in the revised manuscript version. Please find below a point-by-point response:

The authors should provide a more detailed explanation of the molecular subtypes of CRC:

  • A more detailed explanation of the molecular subtypes of CRC was added (see 3.1.5. Clinical, Histopathological, and Molecular Variations in dMMR/MSI-H Colorectal Cancers).

Are there any translational animal models used to study ICI's resistance and to test new approaches to overcome the resistance above:

  • Translational studies investigating approaches to overcome ICI resistance were added (see 5 Conclusion and Prospect).